# Gradual Learning for Semi-Supervised 3D Segmentation

**Johann Christopher Engster**[1,2]          JOHANN.CHRISTOPHER.ENGSTER@IMTE.FRAUNHOFER.DE
**Nele Blum**[1]                                                                      NELE.BLUM@IMTE.FRAUNHOFER.DE
**Laura Hellwege**[1,3]                                         LAURA.HELLWEGE@IMTE.FRAUNHOFER.DE
**Thorsten M. Buzug**[1,3]                                 THORSTEN.BUZUG@IMTE.FRAUNHOFER.DE
**Maik Stille**[1]                                                          MAIK.STILLE@IMTE.FRAUNHOFER.DE

[1] *Fraunhofer IMTE, Fraunhofer Research Institution for Individualized and Cell-Based Medical Engineering, Lübeck, Germany*

[2] *University of Lübeck, Institute for Cardiogenetics, Lübeck, Germany*

[3] *University of Lübeck, Institute of Medical Engineering, Lübeck, Germany*

**Editors:** Under Review for MIDL 2025

## Abstract

We propose an extension of the recently published Gradual Learning, a semi-supervised method for segmentation of slice-stacks. While the original Gradual Learning is based on 2D slices to leverage the high similarity within the local neighborhood, the extension utilizes 3D subvolumes instead. Thus, a 3D segmentation network is trained on initial subvolumes and the corresponding ground truth. Afterward, pseudo labels of the expanded subvolumes are generated, which are reused for training. This process is repeated a set number of times. The approach results in improved segmentation quality without the need for large expert-labeled data sets. The method was evaluated on head magnetic resonance imaging scans for brain segmentation but can be easily transferred to other modalities. The results showed high gains in Intersection over Union scores on a separate test data set (depending on the number of used subvolumes $n$: $n=2$: $0.30 \rightarrow 0.58$, $n=3$: $0.41 \rightarrow 0.63$, $n=10$: $0.55 \rightarrow 0.76$, training with full volume: $0.86$).

**Keywords:** 3D Segmentation, Semi-Supervised Learning, Few-Shot Learning

## 1. Introduction

Artificial intelligence (AI)-assisted segmentation can help clinicians reach a more accurate diagnoses. One downside of most AI methods is their reliance on large datasets with ground truth (GT) masks, which are expensive to generate. To solve this, we proposed a semi-supervised method called Gradual Learning (GL), which showed promising results in medical and industrial applications in magnetic resonance imaging (MRI) (Engster et al., 2024) and computed tomography (CT) (Engster et al., 2025), respectively. GL requires a single hand-labeled 2D slice for the segmentation of 3D slice-stacks, since it relies on high-similarity within the local neighborhood. We now propose an extension of GL that adapts the segmentation process from a 2D-based model trained on 2D slices to a 3D-based model trained on 3D subvolumes.

## 2. Methods

GL employs student-teacher training with pseudo labels (PL), which is similar to other semi-supervised methods. However, during training, no guidance (Lei et al., 2024) is needed since it is assumed that the local neighborhood is inherently similar and learned gradually. An overview of the proposed 3D GL is shown in Figure 1.

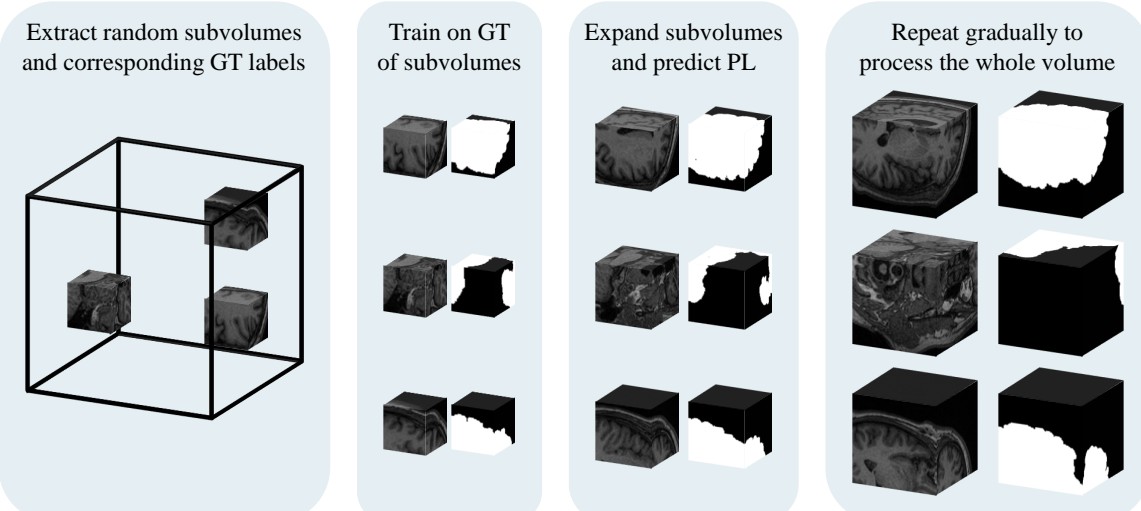

Figure 1: Proposed 3D Gradual Learning, GT = Ground Truth, PL = Pseudo Labels.

The training process involves three key steps. First, a fixed number $n$ of non-overlapping 3D subvolumes of both grayscale values and corresponding GT are randomly extracted from the original 3D volumes. These patches are necessary to capture all relevant anatomical structures during training. Next, a 3D segmentation network is trained on the subvolumes using their GT information. Afterward, the region around each patch is expanded with a small step range $s$ in all dimensions. The trained model is then applied on the expanded subvolume with sliding window inference. Thus, PL are generated for the previously unannotated regions, which are used in the subsequent training. The process is repeated until the entire volume is processed or a number of repetitions is reached. This helps the model to gradually incorporate more contextual information from these neighboring regions.

A 3D U-Net (Çiçek et al., 2016) was selected as segmentation architecture, using the MONAI (Cardoso et al., 2022) library. The initial size of the extracted subvolumes was set to $(80 \times 80 \times 80)$ voxels. The model was then trained on randomly cropped patches of size $(64 \times 64 \times 64)$ within these subvolumes for 500.000 iterations. The step size $s$ was set to eight. The number of GL repetitions was set to ten. For the evaluation, the Calgary-Campinas public dataset (Souza et al., 2017) of MRI head scans for brain matter segmentation was used. The resolution of the scans varied, but was on average $(192 \times 256 \times 256)$. In total, five models were trained on individual scans using GL. Another 25 scans were used for testing.

## 3. Results and Discussion

Figure 2 displays the mean and standard deviation of the average reached Intersection over Union (IoU) on the test data across the five trained models. Different numbers of subvolumes $n = [2, 3, 10]$ were tested. Through the repeated application of GL, the IoU can be increased ($n$=2: $0.30 \to 0.58$, $n$=3: $0.41 \to 0.63$, $n$=10: $0.55 \to 0.76$). This increase is highest in the first $\sim 5$ GL repetitions for all $n$ before stagnating. When using a larger $n$, the reached IoU is higher. For $n = 10$ the standard deviation is highest, likely due to the varying information gain through additional randomly selected subvolumes. The decay in IoU for $n = 10$ after the fifth GL repetition could be due to a deterioration in PL quality. As a benchmark, five models were trained on the full volumes and reached an average IoU of 0.86.

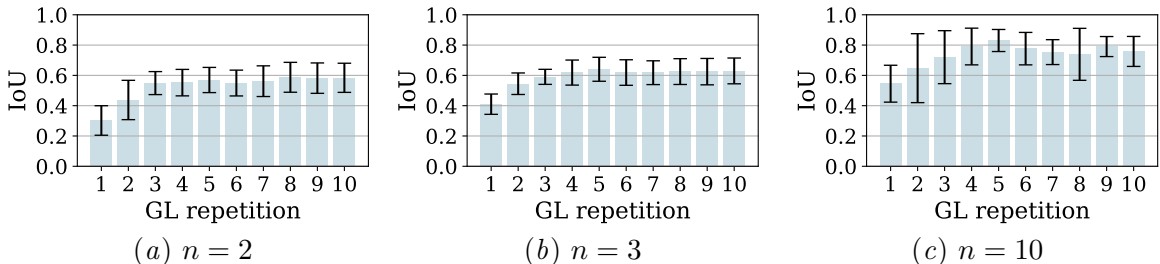

Figure 2: Mean and standard deviation (over all five models trained with GL) of average IoU scores on the test data set. Different numbers of subvolumes $n$ were tested.

Figure 3 shows qualitative examples for test volumes before and after applying GL for $n = [3, 10]$. For $n = 3$ the segmentation before applying GL has many false positive (FP) predicted regions, as most of the head is included. After applying GL, some of these FP regions remain outside the desired brain region. For $n = 10$ the initial segmentation has fewer FP artifacts. After applying GL, the obtained volume appears mostly correct. This shows a clear trade-off between the achieved segmentation quality and necessary subvolumes $n$.

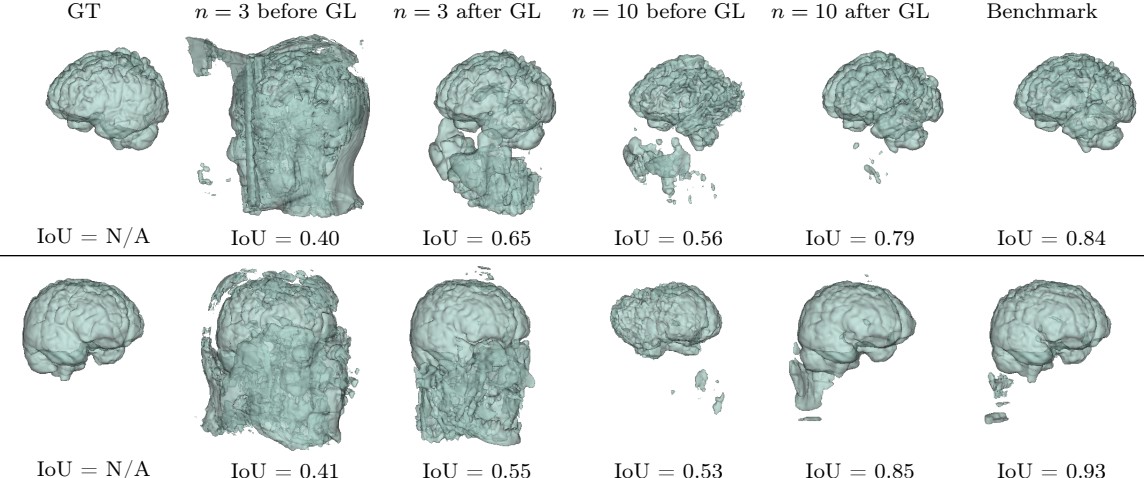

Figure 3: Qualitative examples (test set) showing GT volume and achieved volumes before and after applying GL with different amounts of starting subvolumes $n$. In comparison the benchmark was trained on the full volume instead.

## 4. Conclusion

The results show promising IoU gains through 3D GL for this few-shot learning task. The performance is comparable to 2D GL. However, 2D GL has a high reliance on the correct starting slice and used viewing axis. This makes it difficult to apply for some tasks with complex shapes, which is not the case for 3D GL. As indicated by the varying standard deviation, the placement of the initial subvolumes is also crucial. In the future, artifacts could be removed through post-processing, such as connected component analysis. Similar to 2D GL, 3D GL is not limited to the selected model architecture or imaging modality.

## Acknowledgments

Funded by Land Schleswig-Holstein through the Project "Individualisierte Medizintechnik für bildgestützte, robotische Interventionen (IMTE 2)", Project number: 125 24 009.

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
