# OpenReview forum: "Gradual Learning for Semi-Supervised 3D Segmentation"
_MIDL.io/2025/Short_Papers — MIDL 2025 - Short Papers_

### Official Review · Reviewer_d46R · 2025-04-25

**Rating:** 4
**Confidence:** 4

**Summary:**

This paper proposes a semi-supervised method based on progressive learning for 3D segmentation tasks. Unlike traditional 2D progressive learning approaches, the method leverages 3D sub-volumes instead of 2D slices during training, thereby reducing dependency on the choice of initial slices and viewing angles. According to the authors, this design makes the method more suitable for segmenting objects with complex shapes. The approach begins by training the segmentation network on a small number of 3D sub-volumes with ground-truth labels, then progressively expands the sub-volume range and generates pseudo-labels for subsequent training stages. This training process significantly reduces the need for large amounts of annotated data. Experimental results on a brain tissue segmentation dataset from head MRI scans demonstrate an improvement in IoU after progressive learning, indicating the method's effectiveness to some extent.

**Strengths:**

The proposed method extends conventional 2D progressive learning to 3D, which may partially overcome the limitations of 2D approaches in handling structures with complex shapes.

According to limited experimental results, the method appears to achieve reasonably high-quality segmentation using only a small number of initially labeled sub-volumes.

The authors also claim that the method is agnostic to specific network architectures and imaging modalities, implying potential adaptability to a wide range of medical image segmentation tasks.

**Weaknesses:**

The experimental validation in this paper is limited in several respects:

1.	In the Introduction, the authors state: “To solve this, we proposed a semi-supervised method called Gradual Learning (GL), which showed promising results in medical and industrial applications in magnetic resonance imaging (MRI) (Engster et al., 2024) and computed tomography (CT) (Engster et al., 2025), respectively.” While this suggests that the proposed method has demonstrated effectiveness on both MRI and CT data, the experiments are conducted exclusively on an MRI dataset, with no results provided for CT data. This claim therefore lacks adequate experimental support and weakens the rigor of the presentation.

2.	In the Conclusion, it is emphasized that “Similar to 2D GL, 3D GL is not limited to the selected model architecture or imaging modality,” suggesting that the method is model-agnostic. However, the experiments were conducted using only a single segmentation network, and no validation was performed across different architectures to support this claim.

3.	Key experimental parameters, such as the step size s and the number of GL repetitions, are provided directly without any justification for their selection. The paper does not explore or compare alternative values, leaving it unclear how sensitive the method is to these hyperparameters.

4.	The method is compared only with fully supervised training, lacking direct comparison with other existing semi-supervised 3D segmentation methods. This makes it difficult to comprehensively assess the competitiveness of the proposed approach.

---

### Decision · Program_Chairs · 2025-05-01

Accept